# Targeted Lignan Profiling and Anti-Inflammatory Properties of *Schisandra rubriflora* and *Schisandra chinensis* Extracts

**DOI:** 10.3390/molecules23123103

**Published:** 2018-11-27

**Authors:** Agnieszka Szopa, Michał Dziurka, Angelika Warzecha, Paweł Kubica, Marta Klimek-Szczykutowicz, Halina Ekiert

**Affiliations:** 1Chair and Department of Pharmaceutical Botany, Medical College, ul. Medyczna 9, 30-688 Kraków, Poland; a.warzecha@student.uj.edu.pl (A.W.); p.kubica@uj.edu.pl (P.K.); marta.klimek-szczykutowicz@doctoral.uj.edu.pl (M.K.-S.); mfekiert@cyf-kr.edu.pl (H.E.); 2Polish Academy of Sciences, The Franciszek Górski Institute of Plant Physiology, ul. Niezapominajek 21, 30-239 Kraków, Poland; m.dziurka@ifr-pan.edu.pl

**Keywords:** *Schisandra rubriflora*, *Schisandra chinensis*, red-flowered Chinese magnolia vine, Chinese magnolia vine, lignans, phytochemical analysis, UHPLC-MS/MS, anti-inflammatory activity, LOX, COX, sPLA_2_

## Abstract

*Schisandra rubriflora* is a dioecious plant of increasing importance due to its lignan composition, and therefore, possible therapeutic properties. The aim of the work was lignan profiling of fruits, leaves and shoots of female (F) and male (M) plants using UHPLC-MS/MS. Additionally, the anti-inflammatory activity of plant extracts and individual lignans was tested in vitro for the inhibition of 15-lipooxygenase (15-LOX), phospholipases A2 (sPLA_2_), cyclooxygenase 1 and 2 (COX-1; COX-2) enzyme activities. The extracts of fruits, leaves and shoots of the pharmacopoeial species, *S. chinensis*, were tested for comparison. Twenty-four lignans were monitored. Lignan contents in *S. rubriflora* fruit extracts amounted to 1055.65 mg/100 g DW and the dominant compounds included schisanhenol, aneloylgomisin H, schisantherin B, schisandrin A, gomisin O, angeloylgomisin O and gomisin G. The content of lignan in leaf extracts was 853.33 (F) and 1106.80 (M) mg/100 g DW. Shoot extracts were poorer in lignans—559.97 (F) and 384.80 (M) mg/100 g DW. Schisantherin B, schisantherin A, 6-*O*-benzoylgomisin O and angeloylgomisin H were the dominant compounds in leaf and shoot extracts. The total content of detected lignans in *S. chinensis* fruit, leaf and shoot extracts was: 1686.95, 433.59 and 313.83 mg/100 g DW, respectively. Gomisin N, schisandrin A, schisandrin, gomisin D, schisantherin B, gomisin A, angeloylgomisin H and gomisin J were the dominant lignans in *S. chinensis* fruit extracts were. The results of anti-inflammatory assays revealed higher activity of *S. rubriflora* extracts. Individual lignans showed significant inhibitory activity against 15-LOX, COX-1 and COX-2 enzymes.

## 1. Introduction

*Schisandra rubriflora* (Franch.) Rehd. et Wils, is a rare and little-known plant species of the genus *Schisandra* beyond East Asian phytotherapy. *S. rubriflora* occurs at natural sites in the western Sichuan province of China. It is an endemic species that occurs only in this region [1,2]. *S. rubriflora* cultivations outside the East Asian region are rare, but attempts have recently been made to grow this species in Europe, including Poland [2,3].

*S. rubriflora* is a dioecious vine reaching about 3–4 meters in height [1]. *S. rubriflora* leaves are characterized by elliptical to obovate-elliptical shape, 7–11.5 cm long and 2.5–5.5 cm wide. The leaves are sharp-edged, rarely blunt-edged, and leaf blade edges are finely serrated. Mature berry-shaped fruits of *S. rubriflora*, collected in the hanging ears, are dark red in color, the size of peas, sitting on about 5–8 cm long stalks [1]. 

*Schisandra chinensis* (Turcz.) Baill. is a related species, much better known in terms of medicinal properties, for which cultivation methods have been developed (with cultivations in Europe and America) [1,4,5]. The description of the raw material, i.e., the fruit of the Chinese magnolia vine—*Schisandrae chinensis fructus*—appeared for the first time in 2008 in the European Pharmacopoeia 6 [6]. The raw material has been used for many years in the official health care of Asian countries [7,8,9,10]. It is a pharmacopoeial species also known in the USA [11]. A World Health Organization (WHO) monograph is also devoted to this plant [12]. *Schisandrae chinensis* fruit extracts show valuable, proven, therapeutic properties. These include: anti-inflammatory, anti-tumor, and anti-ulcer properties, anti-bacterial and anti-fungal activity; additionally, they can act hepatoprotectively, adaptogenically and ergogenically; these extract also have antioxidant and detoxification properties [4,13,14].

Scientific information about therapeutic properties of *S. rubriflora* fruits is less available, and its monograph is not listed in any of the world pharmacopoeias [2]. This species is known in the traditional Chinese medicine as a sedative and toning agent, and its fruits are still consumed locally. There are indications regarding the use of this species in the treatment of hepatitis, chronic gastroenteritis and neurasthenia [2,15]. The biological activity of compounds contained in the fruit of this species, described only by Chinese research groups, is limited mainly to the anti-HIV-1 properties, resulting from the inhibition of HIV-1 replication in H9 lymphocytes [16,17]. According to available sources, compounds belonging to the group of dibenzocyclooctadiene lignans as well as nortriterpenoids and bisnortriterpenoids are both responsible for anti-HIV-1 activity [16,18,19]. Furthermore, extracts from *S. rubriflora* shoots have been shown to effectively reduce the level of GPT (glutamin-pyruvate transaminase) in the blood, which may be useful in the treatment of liver and bile duct diseases [2,20].

Valuable biological properties and therapeutic applications resulting from them are conditioned by the unique chemical composition of *S. chinensis* [4,21]. Lignans are the main group of secondary metabolites specific to this genus, among which the main role is played by dibenzocyclooctadiene lignans [21,22]. The majority of scientific research has focused on this group of metabolites. They are referred to as “schisandra lignans” due to the characteristic, complicated chemical structure of these compounds as well as the occurrence limited only to this genus. Schisandrin, gomisin A, deoxyschisandrin and schisantherin A and B are listed as the most important from the group of dibenzocyclooctadiene lignans (Figure 1). Recent studies have reported the identification of ever new structures from the group of lignans and their derivatives [22,23,24]. The available data show that dibenzocyclooctadiene lignans, and their derivatives, specific only for *S. rubriflora*, such as schirubrin A-D, rubrilignans A and B or rubrisandrin A and B are also present in *S. rubriflora* [16,18,25].

There are several studies on the anti-inflammatory activity of *S. chinensis* fruit extracts [26,27] and some individual lignans [28,29,30,31], but there are no studies on *S. chinensis* leaf and shoot extracts. importantly, these investigations have not yet been performed in *S. rubriflora* species. Moreover, there are no studies comparing the results obtained for complex plant material to the results obtained for pure lignans. In this work, we attempt to assess the anti-inflammatory potential of plant extracts and compare it with anti-inflammatory properties of pure lignan samples.

The present study introduces phytochemical characteristics of lignan contents using the UHPLC-MS/MS method in *S. rubriflora*, including the division of the material into female (F) and male (M) specimens of soil-grown plants. The results were compared to the analyses of pharmacopoeial species–*S. chinensis*—performed for comparison purposes. 

Moreover, the anti-inflammatory potential of *S. rubriflora* fruits, leaves and shoots of F and M specimens was studied for the first time using estimations based on the inhibition of eicosanoid-generating enzymes; these included cyclooxygenases (COX-1 and COX-2), lipoxygenase (LOX) and secretory phospholipase A_2_ (sPLA_2_), reducing the concentrations of prostanoids and leukotrienes. Additionally, the analyses involved individual lignans as well as an artificially created “average sample of *S. rubriflora* lignan composition”. Comparative studies with *S. chinensis* shoot, leaf and fruit extracts were also conducted in this study. 

## 2. Results 

### 2.1. Schisandra Rubriflora Lignan Profile

The UHPLC-MS/MS method was used for both qualitative and quantitative analyses of the extracts tested (Appendix A). Twenty-four lignans were quantified in all analyzed samples, representing four lignan groups: dibenzocyclooctadiene lignans (schisantherin A and B, schisandrin, schisandrin C, gomisin A, D, G, J, N, O, 6-*O*-benzoylgomisin O, schisandrin A, rubrisandrin A, epigomisin O, schisanhenol, rubriflorin A, angeloylgomisin H and O), aryltetralin lignan (wulignan A_1_), dibenzylbutane lignans (pregomisin, mesodihydroguaiaretic acid), tetrahydrofuran lignan (fragransin A_2_) and dihydrobenzofuran neolignans (licarin A and B) [32,33,34].

The total lignan content in the analyzed fruit extracts of *S. rubriflora* was 1055.65 mg/100 g DW. Quantitatively dominant compounds were: Schisanhenol (268.02 mg/100 g DW), aneloylgomisin H (185.10 mg/100 g DW), schisantherin B (118.07 mg/100 g DW), schisandrin A (104.32 mg/100 g DW), gomisin O (103.64 mg/100 g DW), angeloylgomisin O (76.88 mg/100 g DW) and gomisin G (66.39 mg/100 g DW) (Table 1).

The presence of twenty-four and twenty-three lignans was found in the analyzed extracts of female (F) and male (M) *S. rubriflora* specimens, respectively (Table 1). No fragransin A_2_ was found in leaf extracts of male specimens. The total contents of the tested group of compounds in leaf extracts were: F—853.33 mg/100 g DW and M—1106.80 mg/100 g DW. 

Qualitatively dominant compounds in F leaf extracts were: Schisantherin B (291.47 mg/100 g DW), schisantherin A (226.80 mg/100 g DW), 6-*O*-benzoylgomisin O (134.51 mg/100 g DW) and angeloylgomisin H (100.83 mg/100 g DW). Qualitatively dominant in M leaf extracts were: 6-*O*-benzoylgomisin O (564.62 mg/100 g DW), angeloylgomisin H (129.28 mg/100 g DW), gomisin D (116.51 mg/100 g DW), schisantherin A (107.17 mg/100 g DW), schisantherin B (104.28 mg/100 g DW) and angeloylgomisin O (48.80 mg/100 g DW) (Table 1).

The presence of twenty-four and twenty-three lignans were found in the analyzed shoot extracts of female (F) and male (M) *S. rubriflora* specimens, respectively. These were the same compounds that were identified in fruit and leaf extracts (Table 1). The total contents of the tested group of compounds in shoot extracts were: F—559.97 mg/100 g DW and M—384.80 mg/100 g DW.

Qualitatively the dominant compounds in F shoot extracts were: Schisantherin B (239.11 mg/100 g DW), angeloylgomisin H (105.80 mg/100 g DW), schisantherin A (84.35 mg/100 g DW) and 6-O-benzoylgomisin O (72.38 mg/100 g DW). Fragransin A_2_ was not found in these extracts. Qualitatively dominant compounds in M shoot extracts were: schisantherin B (169.04 mg/100 g DW), angeloylgomisin H (74.73 mg/100 g DW) and 6-O-benzoylgomisin O (52.18 mg/100 g DW) (Table 1).

### 2.2. Schisandra Chinensis Lignan Profile

The UHPLC-MS/MS analysis of lignans in fruit, leaf and shoot extracts of *Schisandra chinensis* was performed for comparative purposes. When comparing the results, qualitative similarities and quantitative differences were found between the extracts tested (Table 2). In all analyzed samples, twenty-four lignans were quantified, representing four lignan groups: dibenzocyclooctadiene lignans (schisantherin A and B, schisandrin, schisandrin C, gomisin A, D, G, J, N, O, 6-*O*-benzoylgomisin O, schisandrin A, rubrisandrin A, epigomisin O, schisanhenol, rubriflorin A, angeloylgomisin H and O), aryltetralin lignan (wulignan A_1_), dibenzylbutane lignans (pregomisin, mesodihydroguaiaretic acid) and tetrahydrofuran lignan (fragransin A_2_). In addition, dihydrobenzofuran neolignans (licarin A and B) were also found in the analyzed extracts (Table 2). 

The total contents of detected lignans in fruit, leaf and shoot extracts were equal to: 1686.95, 433.59 and 313.83 mg/100 g DW, respectively. Qualitatively the dominant compounds in *S. chinensis* fruits were: gomisin N (259.05 mg/100 g DW), schisandrin A (212.50 mg/100 g DW), schisandrin (206.08 mg/100 g DW), gomisin D (195.22 mg/100 g DW), schisantherin B (195.82 mg/100 g DW), gomisin A (177.94 mg/100 g DW), angeloylgomisin H (161.90 mg/100 g DW) and gomisin J (142.35 mg/100 g DW). Fragrasin A_2_ was not detected in the fruit extract (Table 2).

The amounts of individual compounds were lower in leaf and shoot extracts than in fruit extracts. Rubriflorin A was detected only in trace amounts in the leaf extract. The dominant lignans in both leaf and shoot extracts were: Schisantherin B, gomisin A, gomisin N and angeloylgomisin H, and their quantities were equal to 102.47 and 35.27; 73.82 and 36.29; 55.06 and 62.69; and 47.34 and 44.84 mg/100 g DW, respectively (Table 2). 

### 2.3. Anti-Inflammatory Activity

The following plant material extracts were tested for anti-inflammatory activity: Fruits and leaves of *Schisandra rubriflora* and *Schisandra chinensis* as well as selected most abundant lignans present in plant samples: 6-*O*-benzoylgomisin O, schisandrin, gomisin D, gomisin N and schisantherin A. Extracts from the shoots were not assayed for their anti-inflammatory activity, due to the relatively low lignan contents, determined in phytochemical studies, compared to leaf and fruit extracts (Table 1 and Table 2).

The tests were based on the in vitro inhibition of 15-lipooxygenase (15-LOX), phospholipase A_2_ (sPLA_2_), cyclooxygenase-1 (COX-1) and cyclooxygenase-2 (COX-2) enzymes. 

Plant material extracts showed moderate inhibition of 15-LOX and relatively high inhibitory activity against COX-1, COX-2 and sPLA_2_ (Table 3). Evaluation of 15-LOX inhibition showed that *S. rubriflora* fruit and leaf extracts moderately inhibited this enzyme: 22%—fruits, 38%—F leaves, 42%—M leaves at 17.5 μg/mL. For *S. chinensis*, the activity was lower: 25%—fruits (17.5 μg/mL) and 31%—leaves (175.0 μg/mL) (Table 3).

The sPLA_2_ enzyme inhibition assay showed that fruit and leaf extracts of *S. rubriflora* inhibited its activity to about 62–65% at 175.0 μg/mL. Inhibition percentage for fruit and leaf extracts of *S. chinensis* was lower: 25% and 49%, respectively (at 175.0 μg/mL) (Table 3).

The most promising results were obtained for in vitro inhibitory COX-1 and COX-2 enzyme activities. *S. rubriflora* fruit extracts (at 17.4 μg/mL) inhibited COX-1 and COX-2 activities in 71% and 48%, respectively. Leaf extracts showed higher activity at 175.0 μg/mL, and the inhibition was 86% and 82% (F), and 96% and 90% (M), respectively (Table 3). *S. chinensis* extracts exhibited lower activity. The percentage of COX-1 and COX-2 inhibition was 59% and 66% for fruits, and 69% and 77% for leaves, respectively (Table 3). 

Evaluation of anti-inflammatory properties of individual lignan solutions and the average sample of lignan composition (MIX) (Appendix A) showed that they were not active against the sPLA_2_ enzyme (Table 4). All studied lignans, i.e., 6-*O*-benzoylgomisin O, schisandrin, gomisin D, gomisin N and schisantherin A, as well as their MIX sample, showed from 49% to 57% 15-LOX inhibitory activity at 0.175 μg/mL (Table 4). The highest inhibition for COX-1 was estimated for schisandrin—62% at 1.75 μg/mL, schisantherin A—74% at 0.175 μg/mL, and for the average sample of lignan composition—61% at 1.75 μg/mL (Table 4). The highest inhibition for COX-2 was detected for schisandrin—54% at 1.75 μg/mL, gomisin D—62% at 1.75 μg/mL, gomisin N—70% at 0.175 μg/mL and for the MIX sample—56% at 0.175 μg/mL (Table 4).

## 3. Discussion

Twenty-four lignans were identified from four chemical lignan groups in all analyzed samples of both plant species: Dibenzocyclooctadiene lignans (schisantherin A and B, schisandrin, schisandrin C, gomisin A, D, G, J, N, O, 6-*O*-benzoylgomisin O, schisandrin A, rubrisandrin A, epigomisin O, schisanhenol, rubriflorin A, angeloylgomisin H and O), aryltetralin lignan (wulignan A_1_), dibenzylbutane lignans (pregomisin, mesodihydroguaiaretic acid), and tetrahydrofuran lignan (fragransin A_2_). In addition, the presence of dihydrobenzofuran neolignans (licarin A and B) was also found in the analyzed extracts. Until now, there have been no reports on the detection of so many lignan compounds in *S. rubriflora* fruit, leaf and shoot extracts, including the differentiation on male and female specimens (Table 1, Appendix A).

Schisanhenol was quantitatively predominant in the analyzed *S. rubriflora* fruit extracts (268.02 mg/100 g DW), and its content was: 1.45-, 2.27-, 2.57-, 2.59-, 3.48- and 4.04-fold higher, respectively, than the content of the remaining dominant compounds: Angeloylgomisin H, schisantherin B, schisandrin A, gomisin O, angeloylgomisin O and gomisin G (Table 1).

Twenty-four and twenty-three lignans were found in both leaf and shoot extracts of F and M *S. rubriflora* specimens, respectively. Fragransin A_2_ was found in the extracts from leaves and shoots of F specimens, while it was not detected in analogous extracts from M specimens. The total content of lignans in leaf extracts of F specimens was 1.30-fold lower compared to the content in leaf extracts of M specimens (Table 1). The following compounds were predominant in F specimen leaf extracts: schisantherin A and B, 6-*O*-benzoylgomisin O, and angeloylgomisin H. The contents of schisantherin A and B were: 2.12- and 2.80-fold higher, respectively, compared to their contents in leaf extracts of M. specimens. The contents of 6-*O*-benzoylgomisin O and angeloylgomisin H in leaf extracts of M specimens were: 4.20-, 1.28-fold higher, respectively, than in leaf extracts of F specimens (Table 1). The most dominant compounds in leaf extracts of M specimens were: 6-*O*-benzoylgomisin O, angeloylgomisin H, gomisin D, angeloylgomisin O, schisantherin A and B. Gomisin D and angeloylgomisin O contents in leaf extracts of M. specimens were: 7.08-, 2.84-fold higher, respectively, than in leaf extracts of F specimens (Table 1).

The total lignan content in shoot extracts of F specimens was 1.46-fold higher than in the extracts from M specimens (Table 1). The quantitatively dominant compounds in shoot extracts of F specimens were: Schisantherin B, angeloylgomisin H, schisantherin A, and 6-*O*-benzoylgomisin O: 1.41-, 1.42-, 3.48- and 1,39-fold higher, respectively, compared to shoot extracts of M specimens. Schisantherin B, angeloylgomisin H, and 6-*O*-benzoylgomisin O were predominant in the extracts of M specimens (Table 1).

This work presents for the first time such a wide determination of lignan contents in the extracts of *S. rubriflora*, taking into account the division into the material originating from female (F) and male (M) specimens (leaves and shoots and fruits). Studies on lignan composition of *S. rubriflora* were performed before only by Chinese teams [16,17,18,25,35]. Importantly, these studies were only qualitative, no quantitative data were found, and the authors did not distinguish extracts in terms of gender. In 2006, Chen et al. [18] isolated fruit extracts of *S. rubriflora* and detected following compounds by the ^1^H NMR method: schisanhenol, schisandrin, deoxyschisandrin, schisantherin B, angeloylgomisin P, tigloylgomisin P, gomisin M_1_, M_2_, O and J as well as specific rubrisandrins A and B. In 2010, Xiao et al. [17] identified the following lignans in fruit extracts based on the ^1^H and ^2^H-NMR spectroscopy: Gomisin G and O, angeloylgomisin P, wulignan A_2_, epiwulignan A_1_ and rubrisandrin C. Mu et al. [16] conducted in 2011 an isolation and structure elucidation of the following lignans from *S. rubriflora* fruit extracts using preparative HPLC and ^13^C-NMR: Schisandrin, schisandrin A and C, rubschisantherin, angeloygomisin Q, benzoylgomisin Q, gomisin J, Q, C, B, K, N, S, T, isogomisin O, wilsonilignangomisin G, marlignan L and G. Moreover, the latter authors detected for the first time two new lignans in the fruit extract, i.e., rubrilignans A and B. In comparison to those results, the current study identified eight additional new compounds in fruit extracts: Gomisin A, 6-*O*-benzoylgomisin O, angeloylgomisin H and O, pregomisin, mesodihydroguaiaretic acid and licarin A and B. 

Extracts from aerial parts of *S. rubriflora* were studied by Li et al. [25] using ^1^H- and ^13^C-NMR and they found the following compounds: gomisin K, M_1_ and R, dimethylgomisin J, angeloylgomisin K_3_ and R, interiotherin B, schisantherin D, mesodihydroguaiaretic acid, dihydroguaiaretic acid and pregomisin. Li et al. [35] detected rubriflorin A and B in stem extracts. In the present study, additional nineteen compounds were detected both in shoot and leaf extracts: Schisantherin A and B, schisandrin, schisandrin C, gomisin A, D, G, J, N, O, 6-*O*-benzoylgomisin O, schisandrin A, rubrisandrin A, epigomisin O, schisanhenol, angeloylgomisin H and O, wulignan A_1_ and fragransin A_2_. Moreover, our study included quantitative estimation and division on female and male plants, which had not been done before (Table 1). 

A comparative lignan profiling of extracts from fruits, leaves and shoots of *S. chinensis* using the UHPLC-MS MS method (Table 2 and Table 5) was carried out in the present study. The total content of the tested group of compounds in *S. rubriflora* fruit extracts was 1.60-fold lower than in *S. chinensis* fruit extracts. The total lignan contents in *S. rubriflora* fruit extracts of F specimen was: 1.97-, 2.55-fold higher, respectively, than in *S. chinensis* leaf extracts (Table 1). The total lignan content in extracts from F and M shoots of *S. rubriflora* was: 1.78- and 1.23-fold higher, respectively, than in *S. chinensis* shoot extracts (Table 1).

Schisanhenol, schisantherin B, schisandrin A, gomisin O and angeloylgomisin H were predominant in *S. rubriflora* fruit extracts in terms of quantity (Table 3). The contents of schisanhenol, gomisin O and angeloylgomisin H in *S. rubriflora* fruit extracts was: 27.92- and 1.14-fold higher, respectively, than in *S. chinensis* fruit extracts (Table 1). The total content of schisantherin B and schisandrin A in *S. rubriflora* fruit extracts was: 1.57- and 2.04-fold lower, respectively, than in *S. chinensis* fruit extracts (Table 1).

Schisantherin A and schisantherin B, 6-*O*-benzoylgomisin O and angeloylgomisin H were among the quantitatively predominant compounds in *S. rubriflora* leaf extracts (Table 2). The quantities of these compounds in F and M leaf extracts of *S. rubriflora* were: 58.76- and 27.76-; 2.84- and 1.02-; 12.42- and 52.13-; and 2.13- and 2.73-fold higher, respectively than in *S. chinensis* leaf extracts (Table 5).

Similarly as in leaf extracts, schisantherin A and schisantherin B, 6-*O*-benzoylgomisin O and angeloylgomisin H were among the quantitatively predominant compounds in *S. rubriflora* shoot extracts (Table 2). The quantities of these compounds in F and M leaf extracts of *S. rubriflora* were: 38.00- and 10.93-; 6.78- and 9.68-; 6.98- and 52.13-; and 2.36- and 1.67-fold higher, respectively, than in *S. chinensis* shoot extracts (Table 5).

Different compounds were proved to be dominant in *S. chinensis* extracts. The quantities of schisandrin, gomisin D, gomisin J, gomisin A, schisandrin A, gomisin N and schisandrin C in *S. chinensis* fruit extracts were: 31.37-; 55.46-; 26.36-; 157.25-; 2.04-; 13.49- and 3.47-fold higher, respectively, than in *S. rubriflora* fruit extracts (Table 5). The dominant compounds in *S. chinensis* leaf and shoot extracts included schisandrin, gomisin A, gomisin N and schisandrin C. The amount of these lignans in leaf extracts, in comparison to their quantities in F and M leaf extracts, were: 3.99- and 5.71-; 11.53- and 17.58-; 32.95- and 24.91-; and 84.55- and 195.79-fold higher, respectively. Correspondingly, the quantities of these compounds in shoot extracts, compared to *S. rubriflora* F and M shoot extracts, were: 8.20- and 14.61-; 12.14- and 22.00-; 77.40- and 58.05; and 106.93- and 332.67-fold higher, respectively (Table 5).

The present study determined the complex in vitro anti-inflammatory activity of fruit and leaf extracts of *S. rubriflora* and *S. chinensis*. Moreover, the analyses were also performed on individual most abundant lignans (6-*O*-benzoylgomisin O, schisandrin, gomisin D, gomisin N and schisantherin A) and a synthetic average sample of lignan composition mixture (MIX) (representing mean concentrations of 16 most abundant lignans from fruit and leaf extracts of *S. rubriflora* and *S. chinensis*; see Appendix A). In most cases, dose-dependent inhibition of selected enzymes by plant extracts or lignan solutions was observed. However, the most pronounced exceptions were found for 15-LOX inhibition (and partially COXs), especially by lignan solutions. An inverse dose-dependence in case of plant extracts can be contributed to a relatively wide confidence interval between dilutions (overlapping SDs), and thus the lack of significant difference. However, more interesting is an inverse dose-dependent inhibition of 15-LOX by lignan solutions. This is a clear deviation from the competitive inhibition mechanism. The observed dependencies allow us to suggest the occurrence of another mechanism, namely inhibitor acceleration of the enzyme by lignans. Known mechanisms of inhibitor acceleration rely on allostery and multiple active sites [36]. This assumption could be partially supported by the results for plant extracts, providing also some arguments for significant participation of lignans in their anti-inflammatory properties, albeit of low confidence (as mentioned earlier). This hypothesis should be confirmed in further studies. On the other hand, plant secondary metabolism is so rich and complicated that it is difficult to conclude that particular antioxidant properties, or anti-inflammatory in this case, are driven only by one compound group. Further, a parallel occurrence of components inhibiting as well as increasing the activity is possible due to the natural complexity of such a plant extract.

Previously, there were studies involving anti-inflammatory properties of certain schisandra lignans, but they were conducted on different models. Gomisin N, J and schisandrin C were proved to exert anti-inflammatory effect by reducing nitric oxide (NO) production from lipopolysaccharide-stimulated (LPS) RAW 264.7 cells [28]. Schisantherin A was shown to be an anti-inflammatory agent that down-regulated NF-κB and MAPK signaling pathways in LPS-treated RAW 264.7 cells [29]. Another study [30] demonstrated that schisandrin, deoxyschisandrin, schisandrin B and C and schisantherin A reduced LPS-induced NO production in RAW 264.7 cells. In addition, schisandrin was shown to exert a protective effect on LPS-induced sepsis [31]. In vitro studies performed by Guo et al. [31] showed that anti-inflammatory properties of schisandrin resulted from NO production inhibition, prostaglandin E_2_ (PGE_2_) release, COX-2 and inducible nitric oxide synthase (iNOS) expression, which in turn was caused by the inhibition of nuclear factor kappa B (NF-κB), c-Jun N-terminal kinase (JNK) and p38 mitogen-activated protein kinase (MAPK) activities in the RAW 264.7 macrophage cell line. 

Moreover, extracts from *S. chinensis* fruits were tested for the anti-inflammatory activity by Huyke et al. [26]. Non-polar *S. chinensis* fruit extracts showed that the dose-dependent COX-2 inhibition (at 20 μg/mL) catalyzed prostaglandin production [26]. 

Lim et al. [27] conducted in vitro tests on such representative schisandra lignans as schisandrin, schisandrin A and C, gomisin B, C, G and N, as well as on methanolic extracts of *S. chinensis* fruits, for 5-lipoxygenase (5-LOX) inhibitory activity. The tested compounds inhibited 5-LOX-catalyzed leukotriene production by A23187-treated rat basophilic leukemia (RBL-1) cells at concentrations of 1–100 μM. Compounds, such as schisandrin and gomisins showed moderate inhibitory activity (IC_50_ < 10 μM) against 5-LOX-catalyzed leukotriene production, but they were significantly less active against COX-2-catalyzed PGE_2_ and inducible NO production [27].

We have also proved in our study the high anti-inflammatory potential of *S. chinensis* fruit extracts against COX-1 and COX-2 enzyme activities (Table 3). Positive results were also obtained for leaf extracts (Table 3). Moreover, the inhibitory activity against 15-LOX and sPLA_2_ enzymes has been demonstrated for the first time. To the best of our knowledge, the extracts from fruits and leaves of *S. rubriflora* have not yet been studied for anti-inflammatory activity. The obtained results from *S. rubriflora* plant materials showed a higher activity in comparison to *S. chinensis* (Table 3). 

We have demonstrated, based on the results of individual lignan analyses, that 6-*O*-benzoylgomisin O, schisandrin, gomisin D, gomisin N and schisantherin A display significant 15-LOX, COX-1 and COX-2 inhibitory activities, and that they are virtually inactive against sPLA_2_ (Table 4). Our study also analyzed for the first the anti-inflammatory activity of 6-*O*-benzoylgomisin O and gomisin D. 

## 4. Materials and Methods

### 4.1. Plant Material

Plant material was obtained as part of cooperation with Clematis—Źródło Dobrych Pnączy Spółka z o.o. spółka jawna with a registered office in Pruszków (address: ul. Duchnicka 27, 05-800 Pruszków, Poland) [37]. Plant species were identified by dr. eng. Szczepan Marczyński (owner of the Clematis arboretum). For the purpose of comparative phytochemical analysis, fruits and leaves and shoots (stems with leaves) of about 10 years old female (F) (100 individuals) and male (M) (50 individuals) *S. rubriflora* (Franch.) Rehd. et Wils specimens, and about 10 years old monoecious specimens of *S. chinensis* Turcz. Baill (100 individuals) were collected and dried. Leaves and shoots were harvested in May, fruits in September 2017. The fruits were lyophilized and the leaves and shoots were air-dried (about 25–30 °C). Dry plant material was pulverized in a mixing ball mill (MM 400, Retch, Haan, Germany).

### 4.2. Plant Sample Extraction

Methanol extracts were prepared from fruits, shoots and leaves of F and M *S. rubriflora* plants. The samples (0.3 g, 5 replicates) were extracted with 3 mL of methanol (grade-HPLC, Merck, Darmstadt, Germany). The extraction process was carried out twice in an ultrasonic bath (Sonic 2, POLSONIC Palczyński Sp.J., rsaw, Poland) for 20 min. The obtained extracts were centrifuged for 5 min (4000 rpm) in a centrifuge (Centrifuge MPW–223E, MPW Med. Instruments, Warsaw, Poland). The centrifuged extracts were filtered using sterilizing syringe filters (Millex^®^GP, 0.22 μm, Filter Unit, Millipore, Bedford, MA, USA). 

### 4.3. UHPLC–MS/MS Lignan Targeted Profiling

Lignan-targeted profiling was carried out in methanolic extracts of *S. rubriflora* and *S. chinensis* by means of ultra-high performance liquid chromatography coupled to a tandem mass spectrometer (UHPLC-MS/MS). An external standard addition method was used. Filtered plant extracts were aliquoted in two 45 µL portions. To the first, 5 µL of methanol was added, while to the second 5 µL of the standard lignan solution (all monitored compounds). Samples were analyzed on a UHPLC Infinity 1260 (Agilent, Wolbrom, Germany) coupled to a quadrupole tandem mass spectrometer 6410 QQQ LC/MS (Agilent, Santa Clara, CA, USA). Samples were separated on an analytical column (Kinetex C18 150 × 4.6 mm, 2.7 µm) in a gradient mode of 50% methanol in water (A) versus 100% methanol (B) with 0.1% of formic acid. A linear gradient was applied, 20% to 65% of B in 22 min at 0.5 mL/min at 60 °C; the injection volume was 2 µL. Standard lignan substances were purchased from ChemFaces Biochemical Co. Ltd. (Wuhan, China). The studied lignans and their structures and synonymous names are listed in Appendix A. Lignans were analyzed in the MRM mode after ESI ionization (Appendix A). 

In addition to lignans, whose standards were commercially available (Appendix A), compounds from the dibenzocyclooctadiene lignan group (angeloylgomisins H and O) were also identified based on the UHPLC-MS/MS result analysis for the tested extracts. The identification was based on analyzing fragmentation ions of these compounds visible in mass spectra. The quantitative analysis of angeloylgomisyn H and O was based on their content conversion, according to schisandrin standard curve (UHPLC-MS/MS)—the main compound from the dibenzocyclooctadiene lignan group; according to pharmacopoeial requirements, the raw material should be standardized based on the content of this compound [3].

### 4.4. Anti-Inflammatory Activity

Plant material methanolic extracts of fruits and leaves of *S. rubriflora* (F and M) and *S. chinensis* were tested for anti-inflammatory activity. Additionally, the following most abundant lignans present in the plant samples were analyzed for anti-inflammatory activity: 6-*O*-benzoylgomisin O, schisandrin, gomisin D, gomisin N and schisantherin A (No: L10, L1, L16, L14, L6; Appendix A, respectively); in addition, the mixture of lignans, representing the average plant sample composition (MIX) (mean concentrations of 16 most abundant lignans of fruit and leaf extracts of *S. rubriflora* and *S. chinensis*; see Appendix A), underwent analogous analysis. 

The plant extracts (concentrations: 175.0 and 17.5 μg/mL, Table 3) and solutions of selected lignans (concentrations: 1.75 and 0.175 μg/mL, Table 4) were serially diluted in methanol. The tests were based on in vitro inhibition of 15-lipooxygenase (15-LOX), phospholipases A_2_ (sPLA_2_), cyclooxygenase-1 (COX-1) and cyclooxygenase-2 (COX-2) enzymes.

#### 4.4.1. Inhibitory Activity against 15-Lipooxygenase (15-LOX)

Samples were tested for their inhibitory activity against 15-LOX using an assay kit (760700, Cayman Chem. Co., Ann Arbor, MI, USA), according to the manufacturer’s instructions; arachidonic acid at 0.91 mM was the substrate; nordihydroguaiaretic acid (NDGA) at 100 µM served as a positive control inhibitor. The kit measures the concentration of hydroperoxides produced in the lipooxygenation reaction using purified soy 15-lipooxygenase standard at pH 7.4 in 10 mM Tris-HCl buffer. The reagent’s colorimetric composition is vendor proprietary. The measurements were carried out in 96-well plate using a Synergy II reader (Biotek, Winooski, VT, USA) at 490 nm. The end-point absorbance was recorded after 5-min incubation of enzyme and inhibitor followed by 15-min incubation after substrate addition and 5-min incubation after chromogen addition.

#### 4.4.2. Inhibitory Activity against Cyclooxygenase-1 and Cyclooxygenase-2 (COX-1 and COX-2)

Samples were tested for their ability to inhibit COX-1 and COX-2 using the COX-1 (ovine) and COX-2 (human) inhibitor assay kit (701050, Cayman Chem. Co.), according to the manufacturer’s instructions; arachidonic acid at 1.1 mM was the substrate; ibuprofen at 10 µM served as a positive control inhibitor. The kit measures the peroxidase component of COXs. The appearance of oxidized *N,N,N’,N’*-tetramethyl-*p*-phenylenediamine (TMPD) was monitored kinetically for 5 min in a 96-well plate format at 590 nm using a Synergy II reader.

#### 4.4.3. Inhibitory Activity against Phospholipases A_2_ (sPLA_2_)

Inhibition of sPLA_2_ activity was tested using an assay kit (10004883, Cayman Chem. Co.), according to the manufacturer’s instructions; diheptanoyl thio-PC at 1.44 mM was the substrate; thioetheramide-PC at 100 µM served as a positive control inhibitor. Human recombinant Type V sPLA_2_ was used. Free thiols released by cleavage of the diheptanoyl thio-PC ester bond were measured kinetically using DTNB (5-5’-dithio-bis-(2-nitrobenzoic acid), Ellman’s reagent) in a 96-well plate format at 420 nm using a Synergy II reader. The percent of inhibition was calculated according to Equation (1):%Inh = [(IA−Inhibitor)\IA] × 100(1)
where: %Inh—percent of inhibition; IA—100% enzyme activity (without inhibitor); Inhibitor—enzyme activity with inhibitor added.

All samples were assayed in triplicate, including 100% enzyme activity, positive control inhibitor and tested extracts and lignan solutions.

#### 4.4.4. Statistical Analysis

Quantitative results are expressed in mg/100 g DW (dry weight) as the mean ± SD (standard deviation) of three or five samples (*n* = 3, *n* = 5) in the experiments that were repeated three times.

## 5. Conclusions

The present study is the first comparative, complex, qualitative and quantitative analyses of *S. rubriflora* and *S. chinensis* lignan composition derived from different groups. The contents of shoot and fruit extracts of both plant species were determined for the first time using the UHPLC-MS/MS method. The study identified and characterized twenty-four lignans representing four chemical groups: dibenzocyclooctadiene lignans (schisantherin A and B, schisandrin, schisandrin C, gomisin A, D, G, J, N, O, 6-*O*-benzoylgomisin O, schisandrin A, rubrisandrin A, epigomisin O, schisanhenol, rubriflorin A, angeloylgomisin H and O), aryltetralin lignan (wulignan A_1_), dibenzylbutane lignans (pregomisin, mesodihydroguaiaretic acid), tetrahydrofuran lignan (fragransin A_2_) and dihydrobenzofuran neolignans (licarin A and B). Qualitative and quantitative differences in lignan composition were recorded depending on the origin of samples (fruit, leaf and shoot) as well as plant species.

Additionally, the current work determined for the first time the anti-inflammatory activity, based on the in vitro inhibition of 15-lipooxygenase (15-LOX), phospholipases A_2_ (sPLA_2_), cyclooxygenases 1 and 2 (COX-1; COX-2) enzymes, of fruit and leaf extracts of the analyzed species as well as individual lignans: 6-*O*-benzoylgomisin O, schisandrin, gomisin D, gomisin N and schisantherin A; furthermore, a mixture of lignans representing an average plant sample composition was also tested. The results revealed a high competitiveness of *S. rubriflora* in relation to known, pharmacopoeial plant species—*S. chinensis*.

Based on our research, we suggest to consider the extracts of *S. rubriflora* (fruit, leaf and shoot), as a rich, valuable source of lignans with a promising anti-inflammatory potential. The objects of interest exhibited very interesting differences and showed new research directions involving these compounds, e.g., phenolic composition and other biological activities of *S. rubriflora* would be worth investigating.

## Figures and Tables

**Figure 1 molecules-23-03103-f001:**
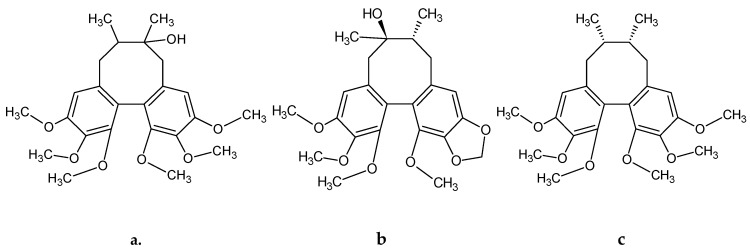
Example structural formulas of abundant *S. rubriflora* dibenzocyclooctadiene lignans: (**a**) schisandrin; (**b**) gomisin A; (**c**) deoxyschisandrin; (**d**) schisantherin A; (**e**) schisantherin B. Structural formulas drown in: ACD/ChemSkech (Freeware), version 12.00, Advanced Chemistry Development, Inc., Toronto, ON, Canada, www.acdlabs.com, 2010.

**Table 1 molecules-23-03103-t001:** The lignan contents [mg/100g DW] ± SD (*n* =5) in fruit and leaf and shoot female (F) and male (M) extracts of *Schisandra rubriflora*.

Lignans	Fruits	Leaves		Shoots	
		F	M	F	M
Wulignan A_1_	19.39 ± 1.35	0.04 ± 0.001	0.04 ± 0.001	0.03 ± 0.002	0.07 ± 0.002
Rubrisandrin A	0.07 ± 0.001	0.06 ± 0.001	0.06 ± 0.001	0.06 ± 0.002	0.10 ± 0.001
Rubriflorin A	traces	traces	traces	traces	traces
Schisandrin	6.57 ± 0.35	8.15 ± 0.25	5.69 ± 0.13	4.01 ± 0.005	2.25 ± 0.01
Gomisin D	3.52 ± 0.16	16.45 ± 0.76	116.51 ± 24.28	8.25 ± 0.42	20.26 ± 0.44
Gomisin J	5.40 ± 0.27	0.97 ± 0.01	0.76 ± 0.03	0.46 ± 0.01	0.36 ± 0.004
Gomisin A	0.75 ± 0.01	6.40 ± 0.18	4.20 ± 0.07	2.99 ± 0.02	1.65 ± 0.01
Gomisin G	66.39 ± 12.37	11.13 ± 1.01	8.23 ± 0.12	5.25 ± 0.04	3.67 ± 0.02
Licarin B	1.98 ± 0.08	0.41 ± 0.06	0.24 ± 0.02	0.19 ± 0.04	0.12 ± 0.01
Epigomisin O	7.46 ± 0.32	10.62 ± 0.41	7.83 ± 0.12	4.91 ± 0.06	3.15 ± 0.01
Gomisin O	103.64 ± 26.25	22.90 ± 1.29	2.81 ± 0.01	12.07 ± 0.20	1.82 ± 0.01
Mesodihydroguaiaretic acid	1.03 ± 0.02	0.34 ± 0.01	0.32 ± 0.004	0.17 ± 0.001	0.16 ± 0.004
Schisantherin A	27.19 ± 3.00	226.80 ± 16.70	107.17 ± 1.66	84.35 ± 4.64	24.27 ± 1.51
Schisantherin B	118.07 ± 18.42	291.47 ± 51.98	104.28 ± 18.16	239.11 ± 38.00	169.04 ± 49.85
Dehydroisoeugenol	0.41 ± 0.004	0.73 ± 0.01	0.44 ± 0.01	0.33 ± 0.004	0.23 ± 0.01
Schisanhenol	268.02 ± 43.12	2.05 ± 0.06	2.73 ± 0.01	1.13 ± 0.003	2.53 ± 0.03
Schisandrin A	104.32 ± 10.11	0.22 ± 0.002	0.38 ± 0.004	0.12 ± 0.001	0.50 ± 0.002
Fragransin A_2_	0.01 ± 0.002	0.01 ± 0.002	Nd *	nd	0.004 ± 0.002
Pregomisin	0.003 ± 0.001	traces	traces	0.01 ± 0.001	traces
Gomisin N	19.20 ± 0.66	1.58 ± 0.02	2.21 ± 0.03	0.81 ± 0.01	1.08 ± 0.01
6-*O*-Benzoylgomisin O	35.28 ± 3.55	134.51 ± 5.91	564.62 ± 33.66	72.38 ± 4.77	52.18 ± 2.63
Schisandrin C	4.96 ± 0.12	0.44 ± 0.03	0.19 ± 0.02	0.28 ± 0.002	0.09 ± 0.01
Angeloylgomisin H	185.10 ± 27.55	100.83 ± 7.89	129.28 ± 13.66	105.80 ± 9.04	74.73 ± 1.54
Angeloylgomisin O	76.88 ± 4.55	17.21 ± 0.26	48.80 ± 3.66	17.25 ± 0.23	26.53 ± 0.46
Total content	1055.65 ± 152.26	853.33 ± 86.85	1106.80 ± 78.33	559.97 ± 57.50	384.80 ± 56.56

Shaded parts indicates the highest quantities of given compounds. *—nd—not detected.

**Table 2 molecules-23-03103-t002:** The lignan contents (mg/100g DW) ± SD (*n* =5) in fruit, leaf and shoot extracts of *Schisandra chinensis*.

Lignans	Fruits	Leaves	Shoots
Wulignan A_1_	0.15 ± 0.03	0.03 ± 0.001	0.04 ± 0.001
Rubrisandrin A	0.03 ± 0.002	0.04 ± 0.001	0.03 ± 0.001
Rubriflorin A	0.01 ± 0.001	traces	0.01 ± 0.001
Schisandrin	206.08 ± 22.32	32.51 ± 3.14	32.87 ± 4.14
Gomisin D	195.22 ± 15.63	9.62 ± 1.96	11.33 ± 1.12
Gomisin J	142.35 ± 19.12	18.06 ± 3.11	13.22 ± 0.54
Gomisin A	177.94 ± 20.14	73.82 ± 8.41	36.29 ± 2.41
Gomisin G	44.56 ± 5.44	12.18 ± 2.14	11.69 ± 1.44
Licarin B	0.37 ± 0.02	0.03 ± 0.001	0.03 ± 0.001
Epigomisin O	3.16 ± 0.09	1.01 ± 0.07	0.91 ± 0.80
Gomisin O	4.08 ± 1.21	5.35 ± 0.55	4.45 ± 0.12
Mesodihydroguaiaretic acid	0.46 ± 0.09	0.38 ± 0.06	0.42 ± 0.07
Schisantherin A	31.32 ± 3.25	3.86 ± 0.98	2.22 ± 0.14
Schisantherin B	185.82 ± 20.39	102.47 ± 4.87	35.27 ± 3.12
Dehydroisoeugenol	0.16 ± 0.05	0.36 ± 0.09	0.41 ± 0.04
Schisanhenol	9.60 ± 1.88	1.00 ± 0.07	0.91 ± 0.02
Schisandrin A	212.50 ± 18.45	17.74 ± 1.02	13.89 ± 1.21
Fragransin A_2_	nd	0.02 ± 0.001	0.01 ± 0.001
Pregomisin	traces	traces	traces
Gomisin N	259.05 ± 30.88	55.06 ± 4.52	62.69 ± 4.98
6-*O*-Benzoylgomisin O	33.64 ± 2.89	10.83 ± 2.01	7.48 ± 1.21
Schisandrin C	18.54 ± 2.15	37.20 ± 2.77	29.94 ± 4.23
Angeloylgomisin H	161.90 ± 15.65	47.34 ± 3.45	44.84 ± 2.27
Angeloylgomisin O	65.56 ± 5.99	4.67 ± 0.87	4.89 ± 0.84
Total content	1686.95 ± 185.67	433.59 ± 40.09	313.83 ± 28.70

Shaded parts indicates the highest quantities of given compounds. nd—not detected.

**Table 3 molecules-23-03103-t003:** In vitro inhibition activity of studied *S. rubriflora* and *S. chinensis* extracts solutions against 15-LOX, COX-1, COX-2, and sPLA_2_.

Plant Extracts		Concentration (μg/mL)	15-LOX		COX-1		COX-2		sPLA_2_	
			% Inh	SD	% Inh	SD	% Inh	SD	% Inh	SD
*Schisandra rubriflora*	Fruits	175.0	1	0.1	32	3.5	52	5.7	62	2.5
		17.5	22	1.5	71	7.8	48	5.2	25	1.0
	Leaves F	175.0	38	2.7	96	10.6	90	9.9	64	2.6
		17.5	42	2.9	51	5.6	40	4.4	54	2.2
	Leaves M	175.0	37	2.6	86	9.5	82	9.0	65	2.7
		17.5	38	2.7	58	6.4	49	5.4	55	2.3
*Schisandra chinensis*	Fruits	175.0	no inhibition	0.0	59	6.5	66	7.3	25	1.0
		17.5	25	1.7	34	3.8	49	5.4	8	0.3
	Leaves	175.0	31	2.2	69	7.6	77	8.4	49	2.0
		17.5	28	2.0	51	5.6	53	5.8	25	1.0
Control inhibitor	NDGA	30.2 (100 μM)	23	2.0	-	-	-	-	-	-
	Ibuprofen	2.1 (10 μM)	-	-	23	2.5	21	2.0	-	-
	Thioetheramide-PC	73.6 (100 μM)	-	-	-	-	-	-	91	4.0

% Inh—percent of enzyme activity inhibition; SD—standard deviation (*n* = 3).

**Table 4 molecules-23-03103-t004:** In vitro inhibition activity of selected pure lignan solutions against 15-LOX, COX-1, COX-2, and sPLA_2_.

Lignans		Concentration (μg/mL)	15-LOX		COX-1		COX-2		sPLA_2_	
			% Inh	SD	% Inh	SD	% Inh	SD	% Inh	SD
6-*O*-Benzoylgomisin O		1.75	18	1.2	35	4	47	5	no inhibition	-
		0.175	49	3.4	19	2	47	5	no inhibition	-
Schisandrin		1.75	31	2.0	62	5.6	54	4.9	no inhibition	-
		0.175	57	4.0	40	4.4	50	5.5	no inhibition	-
Gomisin D		1.75	31	1.8	42	4.2	62	6.4	3	0.1
		0.175	53	3.7	34	3.7	58	6.4	9	0.4
Gomisin N		1.75	16	1.1	42	4.3	58	7.0	6	0.3
		0.175	54	3.8	40	4.4	70	7.6	no inhibition	-
Schisantherin A		1.75	31	1.6	38	5.8	48	5.3	8	0.3
		0.175	55	3.8	74	8.1	31	3.4	1	0.0
Average sample lignan composition (mix)		1.75	20	1.4	61	6.7	43	4.7	no inhibition	-
		0.175	53	3.7	32	3.6	56	6.1	2	0.1
Control inhibitor	NDGA	30.2 (100 μM)	23	2.0	-	-	-	-	-	-
	Ibuprofen	2.1 (10 μM)	-	-	23	2.5	21	2.0	-	-
	Thioetheramide-PC	73.6 (100μM)	-	-	-	-	-	-	91	4.0

% Inh—percent of enzyme activity inhibition; SD—standard deviation (*n* = 3).

**Table 5 molecules-23-03103-t005:** The comparison of the amounts (mg/100g DW) of the dominant lignans in the studied *S. rubriflora* and *S. chinensis* extracts.

Lignans	*Schisandra rubriflora*					*Schisandra chinensis*		
	Fruits	Leaves		Shoots		Fruits	Leaves	Shoots
		F	M	F	M			
Schisandrin	6.57	8.15	5.69	4.01	2.25	206.08	32.51	32.87
Gomisin D	3.52	16.45	116.51	8.25	20.26	195.22	9.62	11.33
Gomisin J	5.40	0.97	0.76	0.46	0.36	142.35	18.06	13.22
Gomisin A	0.75	6.40	4.20	2.99	1.65	177.94	73.82	36.29
Gomisin G	66.39	11.13	8.23	5.25	3.67	44.56	12.18	11.69
Schisantherin A	27.19	226.80	107.17	84.35	24.27	31.32	3.86	2.22
Schisantherin B	118.07	291.47	104.28	239.11	169.04	185.82	102.47	35.27
Schisanhenol	268.02	2.05	2.73	1.13	2.53	9.60	1.00	0.91
Schisandrin A	104.32	0.22	0.38	0.12	0.50	212.50	17.74	13.89
Gomisin N	19.20	1.58	2.21	0.81	1.08	259.05	55.06	62.69
6-*O*-Benzoylgomisin O	35.28	134.51	564.62	72.38	52.18	33.64	10.83	7.48
Schisandrin C	4.96	0.44	0.19	0.28	0.09	18.54	37.20	29.94
Angeloylgomisin H	185.10	100.83	129.28	105.80	74.73	161.90	47.34	44.84
Total content	1055.65	853.33	1106.80	559.97	384.80	1686.95	433.59	313.83

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
