# Peer review of "Targeted Lignan Profiling and Anti-Inflammatory Properties of Schisandra rubriflora and Schisandra chinensis Extracts"

_molecules, 2018, doi:10.3390/molecules23123103_

Reviewer 1 Report

Targeted lignan profiling and anti-inflammatory properties of Schisandra rubriflora and Schisandra chinensis fruit, leaf and shoot extracts, by Szopa et al.

The manuscript focus on two different aspects of two different plant species; 1) quantification of  lignans in extracts of different parts of the plants, and 2) quantification of AIF activity of the same plant extracts. Finally, in order to link the two separate datasets, selected pure lignans are tested for AIF activity.

General comments:

The language in the manuscript need some revision before it can be published. Some phrases and wording are incorrect and should be revised.

Please be consistent in the use of abbreviated family names of the plant species.

The introduction would benefit from a revision as it is a bit hard to read. The authors should also explain why they wanted to investigate the AIF properties of these plants.

Why have the authors chosen to separate between male and female plants? This should be explained.

The quantification of the lignans is based on standards, and structures of all the analyzed  compounds are given in the SI. The authors might consider to show at least one structure in the manuscript itself in order to give the reader an impression of what the molecules look like.

For the AIF screening, the observed activities do not follow a dose-response activity in many cases, i.e. there are higher activities at lower concentrations. This is even the case for the pure compounds. This has serious implications for the interpretation of the results. It is really concerning that the author do not point out this fact or discuss it at all. Do the authors trust their bioactivity data? This must be carefully addressed. The effect of the positive controls should be included in order to assess the potency of the lignins as AIF compounds.

The authors tested the most abundant compounds as well as a mixture of lignans, but they do not compare the activity of the pure compounds to the plant extracts. I assume that this experiment was done in order to investigate wether the lignans could be causing the effect observed in the plants, but this is not discussed at all. Are the authors convinced that the observed AIF effects in the plant extracts are indeed caused by lignans?

The methods of the assays should be outlined in the M&M section even though kits are used.

Specific comments:

TitleSpecies in italics

AbstractSpecies in italics

Table 1What does the gray color in the table indicate?

Table 1Why is there no SD for some of the values? And what is actually the section limit on these analysis? I.e. what is the difference between «nd» and the lowest quantities?

Table 2Same questions as for table 1. And what is the distinction between «traces», «nd» and the lowest quantified amounts?

L148Compared to what?

L159Avoid use of the term «satisfactory»

L161-162The statement «for S. chinensis the activity was lower» is not correct.

L168Replace «at» with «to»

Table 5This table contains only data already presented in table 1 and 2. Presenting identical data several times in the same manuscript should be avoided.

Reviewer 2 Report

      The aim of the paper was the UHPLC-MS/MS 14 targeted lignan profiling of fruits, leaves and shoots of Schisandra rubriflora and Schisandra chinensis extracts. The anti-inflammatory activity of plant extracts as well as individual lignans was tested in vitro for 16 the inhibition of 15-lipooxygenase (15-LOX), phospholipases A2 (sPLA2), cyclooxygenase 1 and 2 17 (COX-1; COX-2) enzyme activities.           

          The manuscript fits within the scope of the journal. The manuscript is interesting and the idea is very nice. The author’s work on discussing achieved results is appreciated. The minor revisions are necessary to improve clarity of the paper. Therefore, I recommend publication of the manuscript with minor revisions.

         I have some recommendations for authors:

1. Please check the full text for typographical errors. For example, use italic style for scientific name of plants (see title and abstract):

- Page 1, Line 13, 19, 20, 25, 27, 29

- Page 1, Line 16: “in vitro” change with “in vitro

2. Please comment in text, why anti-inflammatory tests have not been done on shoots extracts. In the title the authors say: “Targeted lignan profiling and anti-inflammatory properties of Schisandra rubriflora and Schisandra chinensis fruit, leaf and shoot extracts”

3. Please check reference number 35 because in text (Page 10, Line 366) the source was not found.

4. Include in the text potential research directions.
